# Autoxidation Kinetics of Tetrahydrobiopterin—Giving Quinonoid Dihydrobiopterin the Consideration It Deserves

**DOI:** 10.3390/molecules28031267

**Published:** 2023-01-28

**Authors:** Ayoub Boulghobra, Myriam Bonose, Eskandar Alhajji, Antoine Pallandre, Emmanuel Flamand-Roze, Bruno Baudin, Marie-Claude Menet, Fathi Moussa

**Affiliations:** 1Institut de Chimie Physique, CNRS UMR 8000, Université Paris-Saclay, 91405 Orsay, France; 2Faculté de Médecine, Institut du Cerveau et de la Moëlle Épinière, Sorbonne Université, UMR S 1127, Inserm U 1127, UMR CNRS 7225, F-75013 Paris, France; 3Département de Neurologie, Hôpital Pitié-Salpêtrière, AP-HP, F-75013 Paris, France; 4Service de Biochimie, Hôpital A. Trousseau-La Roche Guyon, Assistance Publique—Hôpitaux de Paris, 26, Rue du Dr A. Netter, 75012 Paris, France

**Keywords:** tetrahydrobiopterin, quinonoid dihydrobiopterin, pterins, transient intermediate, hydroxylation, biogenic amines, dopamine, serotonin

## Abstract

In humans, tetrahydrobiopterin (H4Bip) is the cofactor of several essential hydroxylation reactions which dysfunction cause very serious diseases at any age. Hence, the determination of pterins in biological media is of outmost importance in the diagnosis and monitoring of H4Bip deficiency. More than half a century after the discovery of the physiological role of H4Bip and the recent advent of gene therapy for dopamine and serotonin disorders linked to H4Bip deficiency, the quantification of quinonoid dihydrobiopterin (qH2Bip), the transient intermediate of H4Bip, has not been considered yet. This is mainly due to its short half-life, which goes from 0.9 to 5 min according to previous studies. Based on our recent disclosure of the specific MS/MS transition of qH2Bip, here, we developed an efficient HPLC-MS/MS method to achieve the separation of qH2Bip from H4Bip and other oxidation products in less than 3.5 min. The application of this method to the investigation of H4Bip autoxidation kinetics clearly shows that qH2Bip’s half-life is much longer than previously reported, and mostly longer than that of H4Bip, irrespective of the considered experimental conditions. These findings definitely confirm that an accurate method of H4Bip analysis should include the quantification of qH2Bip.

## 1. Introduction

In living organisms, tetrahydrobiopterin (H4Bip) (Figure 1) is the cofactor of several essential hydroxylation reactions [1,2,3]. In humans, H4Bip deficiencies have a serious impact on health, with various pathological conditions occurring at any age, ranging from hyperphenylalaninemia and neurotransmission disorders to cardiovascular diseases, neurodegenerative diseases (Parkinson’s, Alzheimer’s, etc.), depression, inflammatory diseases and cell growth [3]. Nowadays, the pathophysiology of H4Bip remains an active field of research in human medicine, as reflected in the recent description of a new genetic disorder [4], as well as the recent discovery of its role in several pathological conditions, including vitiligo [5,6], autoimmunity and cancer [7], and kidney injury in diabetes [8].

Since the elucidation of the H4Bip structure [2], it has been widely assumed that the first step of the H4Bip-dependent hydroxylation reactions involves the oxidation of H4Bip into a short-lived intermediate quinonoid dihydrobiopterin (qH2Bip) [2,3]. H4Bip is then regenerated by the enzymatic reduction of the latter to dihydro-pteridine reductase (DHPR) (Figure 1) [3]. In the absence of rapid reduction, qH2Bip may rearrange to 7,8-dihydrobiopterin (H2Bip) (Figure 1), a more stable conformer, as reflected in the increase in the latter, in case of DHPR deficiency [9,10,11]. Hence, the determination of these pterins in biological media, including H4Bip, H2Bip and their precursor dihydro-neopterin (H2Nip), is of importance for diagnosis and monitoring the efficacy of H4Bip supplementation in patients with genetic defects [12,13,14]. However, the determination of H4Bip in biological fluids is quite difficult due to its proneness to autoxidation (Figure 1) [15,16,17].

**Figure 1 molecules-28-01267-f001:**
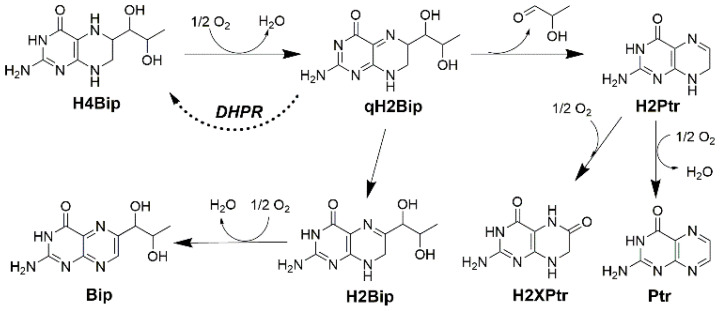
Autoxidation pathways of tetrahydrobiopterin (H4Bip) according to Refs. [2,18,19,20,21,22,23]. qH2Bipquinonoid dihydrobiopterin; H2Bip7,8-dihydro-biopetrin; H2Ptr—dihydro-pterin; H2XPtr—dihydro-xanthopterin; Bip—biopterin; Ptr—pterin; DHPR—dihydro-pteridine reductase.

Under aerobic conditions and in the absence of rapid enzymatic reduction, the first step of H4Bip autoxidation leads to a qH2Bip production [2,18,19,20,21,22] that would then rapidly undergo two competing autoxidation pathways (Figure 1). qH2Bip could either isomerize into H2Bip or lose the side chain at C-6, leading to dihydro-pterin (H2Ptr). Then, H2Bip and H2Ptr would oxidize into biopterin (Bip) and pterin (Ptr) or dihydro-xanthopterin (H2XPtr), respectively [15,16,17,18,19,20,21,22]. These observations underline the central role of qH2Bip in the oxidation pathway of H4Bip and the pathophysiology of H4Bip-dependent hydroxylation reactions. They also suggest that the accurate determination of qH2Bip should provide essential insights in this field.

While the occurrence of qH2Bip in the H4Bip oxidation process has been proposed since 1963 [2], only a few attempts have been made to synthetize [23] or separate and directly characterize this transient intermediate [24,25]. Though strongly suggesting that the first step of H4Bip oxidation led to the release of qH2Bip, these studies, conducted mainly by liquid chromatography coupled to UV–Vis detection, showed that it was very difficult to separate these molecular species [24,25]. Consequently, in the era of the gene therapy of dopamine and serotonin disorders [26], no current method to determine pterins [17,27,28,29,30,31,32,33,34,35,36] allows to quantify qH2Bip. The available data on qH2Bip remain scarce and conflicting [18,19,20]. The autoxidation pathway itself has been sometimes questioned [18,20]. The half-life of qH2Bip would be very short and may vary from 0.9 [24] to 5 min [23] as a function of pH and the type of buffer [20]. Additionally, in biological samples, H4Bip would oxidize within minutes, without a molar relationship between H4Bip loss and the increase in concentrations of H2Bip and Bip [16].

Very recently, by using a multi-analytical approach including differential ion mobility spectrometry (DMS), we have demonstrated that H2Bip isomers can be distinguished and identified based on their MS/MS transition ions [37]. We also reported that the half-life of qH2Bip is longer than it was previously believed under our experimental conditions [37]. However, we did not check the effects of pH and the type of buffer on qH2Bip’s stability. Furthermore, it is worthy to note that, despite the tremendous progress in the field of ion mobility [38,39] and other MS/MS methods [40,41], LC-MS/MS remains more common, notably in medical laboratories where it still constitutes the gold standard in the field of separation techniques. 

Considering the recent findings on the specific detection of qH2Bip [37] based on its characteristic MS/MS transition, it becomes evident that LC-MS/MS may be used to specifically detect this pterin with the multiple reaction monitoring (MRM) mode of detection. Hence, here we developed an HPLC-MS/MS method able to separate all the autoxidation products of H4Bip, including qH2Bip. The main objective is to revisit the autoxidation pathway and kinetics of this essential cofactor as a function of pH and the type of buffer, while providing an LC-MS/MS method potentially applicable to biological media.

## 2. Results

### 2.1. LC-MS/MS Method Development

#### 2.1.1. Optimization of MS/MS Conditions

In order to develop an MS/MS method, we used the product ions scan mode by selecting the [M+H]^+^ parent ion and scanning from *m/z* 50 to *m/z* corresponding to the parent ion for each pterin (Table 1), with collision energies values ranging from 10 to 35 V. For H4Bip, H2Bip, Bip and Nip, the selected precursor, fragment ions and collision energies were optimized by flow injection analysis of 5 µM standard solutions. For qH2Bip, H2Ptr, H2Xptr and Ptr, the acquisition parameters were optimized on a degraded 20 µM H4Bip solution (incubation at room temperature in the dark for 3 h) after HPLC separation. The MRM transitions of the studied pterins are summarized in Table 1.

Except for qH2Bip, we used two MRM transitions for each pterin, the first one corresponding to the most intense fragment for identification, and the second one, less intense, for confirmation. For qH2Bip, we only used the previously deciphered characteristic MRM transition (*m/z*, 240 > 166) [37].

#### 2.1.2. Optimization of Chromatographic Conditions

H4Bip and its oxidation products are low hydrophobic basic compounds, which can be easily separated in hydrophilic interaction liquid chromatography (HILIC) mode or in reversed-phase (RP) mode by using a polar-embedded C18 column, thus avoiding the addition of ion pairing reagents [17,30]. However, HILIC mode separations may suffer from mobile phase/solvent of injection incompatibility for such water-soluble solutes [17]. Hence, we selected a polar-embedded C18 stationary phase, allowing for the separation of pterins in the RP mode [17] with a formate-based mobile phase compatible with the ESI ionization source. For this purpose, we used an Atlantis T3 column with a mixture of pH 2.8, 0.05 M ammonium formate/methanol (97/3, *v*/*v*) as mobile phase, allowing for the protonation of these basic compounds.

#### 2.1.3. Separating the Autoxidation Products

The evolution of the chromatographic profiles of ammonium formate-based WS of H4Bip as a function of pH and incubation time is shown in Figure 2. Under the proposed chromatographic conditions, the MRM mode allows for the specific detection of all targeted autoxidation products of H4Bip, including H2Bip isomers, H2Ptr, Bip, H2Xptr, Ptr and Nip used as internal standard, in less than 3.5 min (Figure 2a–e).

The chromatograms of Figure 2a–e clearly show the separation between qH2Bip and H2Bip, but not a significant separation between qH2Bip and H4Bip. However, irrespective of their separation, it is possible to accurately determine the inseparable molecules with characteristic fragmental ions using advanced tandem MS (Table 1, and Appendix A). Under the proposed chromatographic conditions, despite the co-elution of some autoxidation products, there is a clear resolution (Rs > 1.1) between the critical pair of peaks (i.e., H4Bip and qH2Bip) sharing the ionic fragment *m/z* 166. 

Nevertheless, Figure 2a,b,d show that qH2Bip signals contain a small peak ahead of the major peak that matches the exact elution profile of H4Bip. The major peak is well separated from the H4Bip peak. It corresponds to the specific MS/MS transition of qH2Bip (Appendix A) released from H4Bip during incubation. Indeed, its intensity increases over the incubation time in parallel with the decrease in the intensity of the H4Bip peak (Figure 2a,b). The small peak that matches the retention time of H4Bip (Figure 2a,c,d) can be attributed, without doubt, to the oxidation of some of the H4Bip to qH2Bip in the ionization source, for two reasons: The first reason is that the MS spectrum extracted from the H4Bip peak (Appendix A) does show a fragment ion of m/z 240, corresponding to qH2Bip, whereas the MS/MS spectrum of H4Bip does not (Appendix A). The second reason is that the intensity of this peak decreases in parallel with that of the H4Bip peak (Figure 2a,b), to completely disappear with the disappearance of H4Bip after ten hours of incubation (Figure 2c). The oxidation phenomenon at the electrospray ionization source is well known for easily oxidizable molecules, such as peptides and amino acids, for example [42]. As H4Bip is readily oxidizable, some authors even recommended its stabilization by derivatization with benzoyl chloride before LC-MS/MS analysis [43].

To the best of our knowledge, this is the first time that qH2Bip has been separated from H4Bip and unambiguously identified by HPLC-MS/MS, thus offering the possibility of studying its autoxidation kinetics.

As a stationary phase, we also tried a 100 × 2.1 mm, 1.8 µm HSS T3 column assumed to be the UHPLC counterpart of the Atlantis T3 column. However, we were not able to separate qH2Bip from H4Bip with this stationary phase. The loss of resolution with the HSS T3 column may be due to some differences in chemical composition as compared to the Atlantis T3 column, and/or more pronounced extra-column effects (void volume) for the HSS T3 column, resulting in a loss of efficiency as compared to the Atlantis T3 column.

It is possible to use an Atlantis T3 column with larger dimensions (length and diameter) than that proposed herein in order to increase the efficiency of the separation and, thus, the resolution. However, this will require a longer run-time of analysis, which is not appropriate for the separation of molecules with short half-lives. Furthermore, peak volumes (peak width in volumetric terms) increase with increasing length and column cross-section, or the square of the change in diameter, resulting in proportionally smaller peaks (loss of sensitivity). It is also possible to optimize the mobile phase and elution conditions in order to better separate the autoxidation products. However, the proposed column geometry is enough to separate the critical pair corresponding to the H4Bip and qH2Bip peaks under isocratic elution in a short run-time convenient for autoxidation studies, which are our main objective.

Moreover, the percentage of methanol in the mobile phase is only 3%, which does not favor the yield of ionization in the ESI source. It is therefore possible to improve the ionization efficiency and, thus, the detection sensitivity, by adding some percentage of methanol to the column outlet through a T union. However, this will need an additional pumping system, whilst the sensitivity of the proposed method is enough to detect all oxidation products of H4Bip without the need for any additional material.

### 2.2. Autoxidation Kinetics

In 1963, H2Bip was identified as the stable product of qH2Bip rearrangement in a pH 6.8, 0.1 M phosphate buffer [2]. Twenty years later, this assignment was questioned because the aerobic oxidation of H4Bip in a pH 7.6, 0.1 M Tris buffer mainly led to H2Ptr [18]. To answer these questions, the authors of the former study monitored the rearrangement of qH2Bip spectrophotometrically in various buffer solutions [20]. According to the authors, qH2B half-life values may vary from 1.2 to 2.0 min [20]. The authors also stated that, in a phosphate buffer at pH 6.8, the predominant product of the aerobic oxidation of H4Bip is H2Bip, whereas in a Tris buffer at pH 7.6, the analogous reaction yields H2Ptr [20]. The authors concluded that the pH of the solution is not the only factor that determines which pathway the rearrangement of qH2Bip follows. The temperature and the type of buffer can also play important roles during the rearrangement of qH2Bip in determining the distribution of products [20]. However, all these data were obtained at neutral pH and were only based on the spectrophotometric detection of H4Bip autoxidation products. Moreover, the data related to H4Bip stability and qH2Bip half-life remain scarce and conflicting [23,24,37]. Thus, we used the proposed LC-MS/MS method to further investigate this issue.

#### 2.2.1. Selecting pH and Buffers

In order to investigate the autoxidation pathway of H4Bip’s and qH2Bip’s half-life, we selected three pH levels covering acidic and neutral conditions. We first selected pH 2.8, which is the pH of the mobile phase of the proposed method. This pH is the most suitable one for the protonation and MS/MS detection of these basic compounds, whilst considering the stationary-phase stability under acidic conditions. We also selected pH 5.4 because this pH is commonly used for the RP-LC separation of pterins in biological media [17]. Finally, we chose to investigate the autoxidation of H4Bip at the physiological pH level, i.e., pH 7.4. Additionally, this latter pH is the only condition enabling the fluorescence detection of H4Bip after post-column electro-oxidation [17,30].

In order to investigate the effects of the type of buffer, we selected three ammonium-based buffers. Ammonium formate was selected because it is the main constituent of the proposed mobile phase and the most widely used volatile buffer for HPLC-MS separations [27,28,29,30,31,32,33,34,35,36]. We also selected ammonium acetate because it is frequently used for the analysis of pterins in biological samples [17]. Finally, we selected ammonium citrate because of its good buffer capacity in the selected entire pH range, and because of its chelating properties. Citrate buffer is also used for pterin determination in biological media [30]. For obvious reasons of incompatibility with the ionization source, we avoided using phosphate buffers.

#### 2.2.2. Bip and H2Bip Quantification

As H4BIP would oxidize within minutes without a molar relationship between H4Bip’s loss and the increase in the concentrations of H2Bip and Bip [16], we aimed to quantify both oxidation products under the selected autoxidation conditions. For this purpose, we checked the linearity of the proposed method between 5 and 2000 nM for both pterins (y = 0.006x − 0.021, *n* = 6, and y = 0.005x + 0.057, *n* = 6, with r^2^ > 0.99 for H2Bip and Bip, respectively) (Appendix A). The accuracy of the method was investigated at two concentration levels, 5 and 1000 nM. By using Nip as IS, the relative standard deviations were lower than 5% for both compounds in all instances (Table 2).

#### 2.2.3. Autoxidation at pH ≤ 3.0

The autoxidation kinetics under acidic conditions is shown in Figure 3 (panels a–c). The autoxidation pathway remains the same regardless of the type of buffer. The first autoxidation product is qH2Bip, which converts to H2Bip, which slowly oxidizes to yield Bip. In all instances, H2Ptr and Ptr are very low and remain below the limit of quantitation. Hence, under these conditions, qH2Bip is mainly converted into H2Bip. However, the kinetic parameters significantly differ as a function of the type of buffer (Table 3).

In the formate buffer (Figure 3a), H4Bip decreases bi-exponentially. From 0 to 3.4 h, the mean of the first half-life is 87 min (*n* = 3). The mean of the second half-life (*n* = 3), from 3.4 to 8.7 h, drops to 33 min (Table 3). Under the same conditions, qH2Bip first increases from 0 h to 0.8 h, and then starts decreasing bi-exponentially, similarly to what occurs for H4Bip, to yield H2Bip. Between 0.8 and 3.9 h, the mean of the first half-life of qH2Bip is 111 min (*n* = 3), while the second one, between 3.9 and 9 h, drops to 37 min. The descriptive statistical analysis of the mean half-lives of qH2Bip shows that they are significantly higher than those of H4Bip.

Meanwhile, H2Bip starts increasing from 0 to 4.9 h and then begins decreasing to yield mainly Bip. The quantities of qH2Bip and Bip remaining at the end of the experiment, i.e., 25 h, represent, respectively, 30% and 40% of the starting quantity of H4Bip. At the same time, the decrease in qH2Bip also coincides with a significant increase in H2XPr (Figure 2a), thus suggesting that part of qH2Bip is oxidized into H2XPtr under these conditions. Along with the traces of H2Ptr and Ptr present in the medium, the amount of H2XPtr could explain the 30% deficit recorded. However, this remains to be confirmed by a precise determination of all the oxidation byproducts.

As compared to the formate buffer, an acceleration in the autoxidation kinetics is observed in the acetate buffer (Figure 3b, Table 3). This could explain why the starting relative abundance of H4Bip is widely weaker than that observed for the formate buffer (Figure 3a) while the starting concentration was the same. At time 0, more than 80% of H4Bip is already oxidized into qH2Bip. In contrast with the bi-exponential decrease observed in the formate buffer, H4Bip decreases exponentially between 0 and 4 h, with a half-life of 25 min. Concomitantly, qH2Bip decreases between 0 and 5 h with a half-life of 34 min (Figure 3b, Table 3). Meanwhile, H2Bip increases from 0 h to 5 h and then starts decreasing until the end of the study (25 h). H2Bip seems more stable under these conditions, and the amounts of H2Bip (58%) and Bip (15%) determined after 25 h of incubation are slightly higher than those recorded for the formate buffer. However, the traces of H2XPtr, H2Ptr and Ptr detected do not seem to fill the 27% deficit, as compared to the initial quantity of H4Bip.

In the citrate buffer (Figure 2c), H4Bip also oxidizes rapidly into qH2Bip, similarly to what occurs in the acetate buffer. However, under these conditions, qH2Bip is more stable than in the latter, with a half-life nearly five times longer (Table 3). The H2Bip resulting from the slow conversion of qH2Bip is also very stable under these conditions. At the end of the experiment, the quantity of H2Bip represents 82% of the initial quantity of H4Bip, while Bip only represents 7%. Without a precise dosage, it is difficult to know whether the other detected oxidation products (H2Ptr, Ptr and H2XPtr) can fill the nearly 10% deficit as compared to the initial quantity of H4B.

#### 2.2.4. Autoxidation at pH 7.4

The autoxidation kinetics under neutral conditions is shown in Figure 3 (Panels d–f, Table 3). As compared to acidic conditions, the oxidation pathway and kinetics at neutral pH change drastically (Table 3). 

In the formate buffer, qH2Bip mainly loses its side chain to give H2Ptr. Under these conditions, H2P is highly stable, while the amounts of the other oxidation products remain very low (5% H2Bip, and <1% Bip) at the end of the experiment (Figure 3d). These results are in line with those reported for the chemical oxidation of H4B into Ptr under alkaline conditions (0.1 M NaOH) [15], and likely explain why the chemical oxidation of H4Bip to pterin is not complete after one hour of incubation [17].

At neutral pH in the formate buffer, H4Bip’s autoxidation starts at 0.3 h after a short plateau where less than a 5% of H4Bip decrease is observed. Then, H4Bip decreases bi-exponentially with half-lives significantly longer than those observed under acidic conditions (Table 3). Likewise, qH2Bip decreases bi-exponentially, with a first half-life of 135 min between 0.3 and 6.4 h, and a second half-life of 75 min between 6.4 and 14.8 h (Table 3). Interestingly, the half-lives of H4Bip and qH2Bip are significantly longer than those observed in the acidic media. These results disagree with a previous hypothesis suggesting that qH2Bip’s isomerization into H2Bip is slower than the loss of the lateral chain leading to H2Ptr, and that is why the isomerization only occurs when qH2Bip is stable enough not to be oxidized into H2Ptr [22]. The data obtained herein clearly show that the loss of the side chain of qH2Bip is only dependent on pH, regardless of the stability of this isomer.

In the acetate buffer at neutral pH, the main differences with what occurs in the formate buffer lie in the kinetics of H4Bip’s and qH2Bip’s degradation, as well as in the subsequent oxidation of H2Ptr (Figure 3e). In contrast to what occurs in the formate buffer, a significant portion of qH2Bip converts to H2Bip, while the other oxidation products, notably Ptr and H2Xptr, slowly increase as a function of the incubation time (Figure 3e). Under these conditions, the amounts of H2Bip produced through qH2Bip conversion remain quite stable until the 25th hour (9% of the initial amount of H4Bip), while the amounts of Bip represent only 3%.

In contrast to what happens in both the formate and the acetate buffers at neutral pH levels (Figure 3d, e), the degradation of qH2Bip in the citrate buffer produces a mixture of H2Bip and H2Ptr (Figure 3f). Additionally, the oxidation of H4Bip into qH2Bip is slower, while the half-life of qH2Bip is significantly longer than under acidic conditions (Figure 3c, Table 3). Under these conditions, only H2Bip remains stable, while H2Ptr decreases to mainly yield H2Xptr (Figure 3f). After 25 h of incubation, the amounts of H2Bip and Bip represent 30% and less than 1% of the initial amount of H4Bip, respectively.

#### 2.2.5. Autoxidation at pH 5.4

At this pH, the kinetics of autoxidation are globally faster (Table 3), but, again, the pathway differs as a function of the type of buffer (Figure 3g–i).

In the formate buffer, the breakdown of H4Bip and qH2Bip rapidly yields a mixture of H2Bip and H2Ptr. While the latter rapidly autoxidizes into H2Xptr and Ptr, the former slowly oxidizes to yield Bip (Figure 3g). At the end of the experiment, the amounts of H2Bip and Bip represent 15% and 18% of the initial amount of H4Bip, respectively.

In the acetate buffer, qH2Bip mainly isomerizes into H2Bip, similarly to what occurs at more acidic pH values (Figure 3b), but the latter becomes highly stable (Figure 3h). At the end of the experiment, the amounts of H2Bip and Bip represent 64% and 12% of the initial amount of H4Bip, respectively.

In the citrate buffer, the oxidation pathway (Figure 3i) is quite similar to that of the acetate buffer (Figure 3h), but H2Bip is significantly less stable (Figure 3i). At the end of the experiment, the amounts of H2Bip and Bip represent 34% and 7% of the initial amount of H4Bip, respectively.

### 2.3. Effects of H4Bip Concentration

Although the results obtained by HPLC-MS/MS confirm those obtained by the DMS technique [37] concerning the near equivalence of the half-lives of H4Bip and qH2Bip, the values reported by both methods are significantly different. Indeed, the latter technique shows that both pterins remain present in the medium beyond the 73rd hour in the formate buffer [37]. Since the concentrations of H4Bip used in the DMS study are, for sensitivity reasons, 100 times higher than those used in the present study, we checked the influence of this factor on the half-lives of both molecular species.

The data obtained show that half-lives increase significantly with the increase in H4Bip concentration (Figure 4, and Table 4). This is most likely due to the amount of dissolved oxygen in the medium. A more precise study of the influence of this factor on H4B autoxidation is underway in our laboratory.

## 3. Discussion

Considered together, the results obtained herein with a specific mode of detection mostly confirm the hypothesis concerning the role of the type of buffer [20]. Specifically, the present data clearly indicate that, for a given type of buffer, the kinetics and pathway of autoxidation are strongly dependent on pH (Figure 3, Table 3). Overall, these results show that the autoxidation of H4Bip at acidic pH (i.e., ≤3.0) mainly leads to the production of H2Bip after the isomerization of qH2Bip (Figure 3). Under neutral conditions, however, the loss of the qH2Bip side chain leading to H2Ptr is the most favorable pathway. These observations mostly agree with what is generally observed after chemical oxidation [15]. In the presence of a chelating agent such as citrate ions, however, the degradation of qH2Bip at neutral pH produces a mixture of H2Bip and H2Ptr (Figure 3f).

All these observations still do not explain why it is H2Bip that is predominantly excreted in vivo, i.e., under neutral conditions [10,11]. In this context, it has been previously suggested that qH2Bip’s isomerization into H2Bip is slower than the loss of the lateral chain leading to H2Ptr, and that is why the isomerization only occurs when qH2Bip is stable enough not to be oxidized into H2Ptr [22]. The results of the present study obtained in the formate and acetate buffers showing the stability of qH2Bip at neutral pH values, i.e., longer half-lives than under acidic conditions (Figure 3, Table 3), clearly invalidate this hypothesis. Indeed, despite its stability under neutral conditions, qH2B mainly oxidizes into H2Ptr with a negligible yield of H2Bip (Figure 3d,e). Rather, the present data suggest that the in vivo microenvironment of qH2Bip must be considered to explain why DHPR-deficient patients who cannot reduce qH2Bip to H4Bip excrete H2Bip and not H2Ptr. The in vivo microenvironment of qH2Bip, notably its binding to hydroxylases, could slow or even inhibit the loss of the alkyl side chain, similarly to what partially occurs with citrate ions at neutral pH values (Figure 3f).

Concerning the half-lives of H4Bip and qH2Bip, the results obtained herein (Table 3) obviously deserve to be verified with a larger number of samples. In the meantime, these results confirm those obtained with the DMS technique [37]. It is worthy to note, however, that these half-lives depend on several factors, including pH and the type of buffer, as well as H4Bip concentration and, most likely, temperature, of course. It will be then very tedious and time-consuming to determine these half-lives with enough precision. However, these half-lives are important from an analytical point of view, both for the conservation and the treatment of the sample. It will therefore be necessary to determine them precisely for each operating condition. Meanwhile, the most important information to take away from all these results is that, regardless of the type of buffer, pH level or starting concentration, the half-life of qH2Bip is mostly longer than that of H4Bip.

## 4. Materials and Methods

### 4.1. Chemicals

All standards, including H4Bip, H2Bip, Bip and neopterin (Nip), were purchased from Sigma and were used without further purification. Standard solutions (SS, 1 mM) of H4Bip, H2Bip, Bip and Nip (0.25 mM, used as internal standard (IS)) were prepared in pH 2.8, 0.1 M ammonium formate, aliquoted and immediately stored at −20 °C.

### 4.2. Instrumental Analysis

HPLC-MS/MS analyses were carried out on a Shimadzu Nexera LC-20A system connected to a Shimadzu LCMS-8040 triple quadrupole mass spectrometry.

#### 4.2.1. Liquid Chromatography

Isocratic chromatographic separations were performed on a 100 × 2.1 mm, 3 µm Atlantis T3 C18 column (Waters, St-Quentin en Yvelines, France). The mobile phase consisted of a mixture of pH 2.8, 0.05M, ammonium formate/methanol (97/3; *v*/*v*). The flow rate was set at 0.4 mL/min and the column temperature was maintained at 30 °C.

#### 4.2.2. Mass Spectrometry

MS/MS experiments were performed in multiple reaction monitoring (MRM) acquisition modes with positive electrospray ionization source. ESI source voltage was set at 4.5 kV, heat-block temperature at 400 °C and desolvation temperature at 250 °C. Nitrogen was used as drying gas at a flow rate of 15 L/min. Nitrogen was also used as nebulizing gas at 3 L/min. Argon was used for collision-induced dissociation.

### 4.3. Sample Preparation for Autoxidation Kinetic Studies

Kinetics studies were conducted in three different 0.1 M ammonium-based buffers including formate, acetate and citrate, at three different pH levels each (≤3.0, 5.4 and 7.4).

Working solutions (WS) for HPLC-MS/MS autoxidation kinetics studies are prepared extemporaneously (within 5 min) after appropriate dilution of defrosted aliquots of SS in the considered buffer. The final concentrations of H4Bip and IS in the WS were set at 1.00 and 0.25 µM, respectively. WS are then installed at 10 °C in the autosampler and 10 µL is injected into the chromatograph at appropriate intervals of time.

### 4.4. Autoxidation Reaction Order, Rate Constants and Half-Lives

In order to estimate the autoxidation reaction orders of H4Bip and qH2Bip, we selected the most appropriate function in terms of coefficient of determination (r^2^) to modeling their kinetics of decrease. Depending on whether the model is linear or exponential, the autoxidation reaction order will be 0 or 1, respectively [44,45]. For a linear model, the corresponding equation to calculate the half-life follows Equation 1: A (t) = A_0_ − kt; t_1/2_ = A_0_/2k, but it follows Equation 2: A (t) = A_0_ e^−kt^; t_1/2_ = ln(2)/k for an exponential model, where A(t) is the measured area at t hours, A_0_ is the area at t = 0 h, k is the rate constant and t_1/2_ is the half-life [44,45].

### 4.5. Bip and H2Bip Quantification

In order to quantify Bip and H2Bip release during H4Bip autoxidation, we checked the linearity and the precision of the LC-MS/MS proposed method for both products.

### 4.6. Statistical Analysis

The experiments in formate buffer were performed in triplicates on three different days for both pH 2.8 and pH 7.4. The statistical significance of the mean differences was performed by descriptive statistical analysis on Stata. All *p*-values < 0.05 were considered statistically significant.

## 5. Conclusions

As compared to the data we obtained with the multiple analytical approach [37], the results of the present study obtained by LC-MS/MS specifically show that the half-life of qH2Bip is almost always longer than that of H4Bip, regardless of the experimental conditions. As discussed previously [37], this phenomenon is not surprising, since qH2Bip is the first step in the autoxidation of H4Bip. Thus, as long as there is H4Bip in the medium, there will necessarily be a production of qH2Bip.

Since the half-life of qH2Bip is equivalent to, or even longer than, that of H4Bip, it becomes obvious that an accurate method of H4Bip analysis should consider, henceforth, the quantification of qH2Bip, including the qH2Bip formed in the ESI source (Figure 2a,b). It is simply a matter of giving qH2Bip the consideration it deserves. This is of importance for any study concerning the diagnosis of H4Bip deficiency and the quality control of H4Bip-based drugs, as well as pharmacokinetics studies and the therapeutic monitoring of H4Bip supplementation.

The application of the proposed method to the determination of these pterins in biological media is currently under validation in our laboratory.

## Figures and Tables

**Figure 2 molecules-28-01267-f002:**
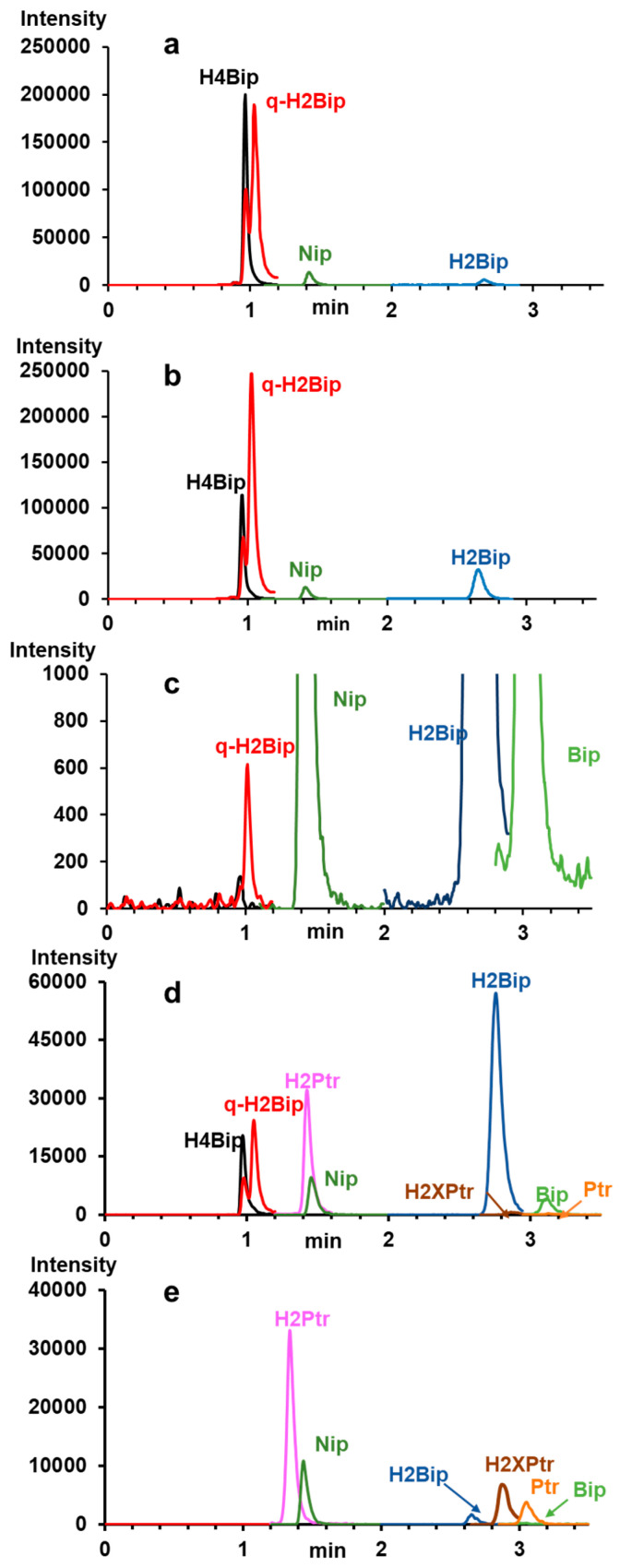
Chromatographic profiles of H4Bip working solutions prepared and incubated in 0.1 M ammonium formate buffer at (**a**) pH 2.8 at time 0; (**b**) pH 2.8, after 1 h of incubation; (**c**) pH 2.8, after 10 h of incubation (zoom-in on the peak of residual qH2Bip); (**d**) pH 5.4 after 1.6 h of incubation; (**e**) pH 7.4 after 49.4 h of incubation. Chromatographic conditions are detailed in the experimental section.

**Figure 3 molecules-28-01267-f003:**
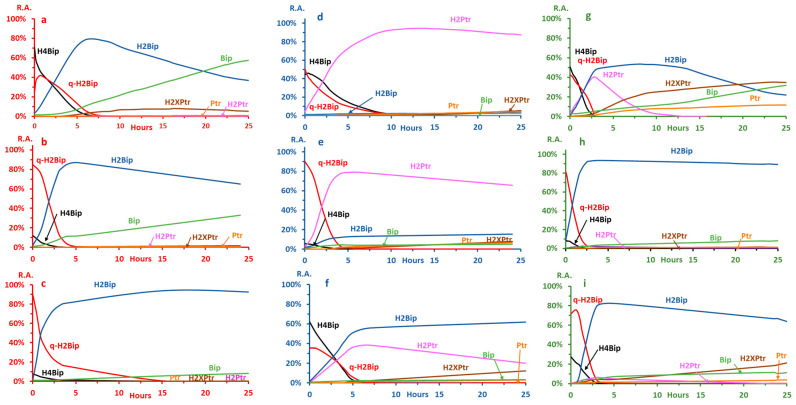
H4Bip autoxidation kinetics as a function of pH and type of buffer (0.1 M ammonium buffer): (**a**–**c**) pH ≤ 3.0; (**d**–**f**) pH 7.4; and (**g**–**i**) pH 5.4. (**a**) pH 2.8, formate; (**b**) pH 3.0, acetate; (**c**) pH 2.8, citrate; (**d**) pH 7.4, formate; (**e**) pH 7.4, acetate; (**f**) pH 7.4, citrate; (**g**) pH 5.4, formate; (**h**) pH 5.4, acetate; and (**i**) pH 5.4, citrate. R.A.—Relative abundance. R.A. is expressed as the percentage of each peak-area ratio (PAR) (PAR—peak area of the considered product/IS peak area) relative to the total sum of all PARs.

**Figure 4 molecules-28-01267-f004:**
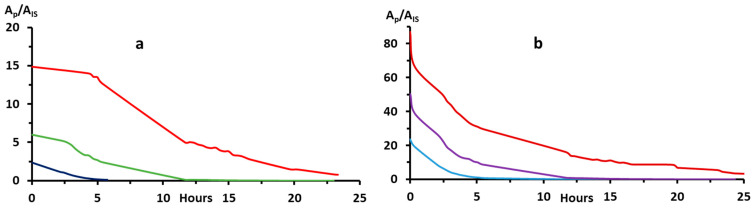
Autoxidation kinetics as a function of H4Bip concentration: (**a**) H4Bip autoxidation (0.5, 1.0, and 2.0 µM); (**b**) qH2Bip degradation.

**Table 1 molecules-28-01267-t001:** MS/MS transitions of targeted pterins (most intense fragment in bold). H4Bip (Tetra-hydro-biopterin), qH2Bip (Quinonoid dihydro--biopterin), H2Ptr (Dihydro-pterin), N (Neopterin), H2Bip (7,8-Dihydrobiopetrin), H2XPtr (di-hydroxy-xanthopterin), Bip (biopterin), Ptr (Pterin).

		MRM Transitions				
Analyte	Retention Time (min)	Parent Ion (*m/z*)	Fragment Ions (*m/z*)	Q1 Potential (V)	Collision Energy (V)	Q3 Potential (V)
H4Bip	0.93	242	**166**	−27.0	−20.0	−30.0
			206	−13.0	−18.0	−21.0
qH2Bip	1.05	240	**166**	−27.0	−15.0	−16.0
H2Ptr	1.43	166	**107**	−19.0	−23.0	−11.0
			121	−18.0	−21.0	−20.0
N	1.46	254	**206**	−29.0	−16.0	−25.0
			236	−29.0	−19.0	−21.0
H2Bip	2.76	240	**196**	−27.0	−14.0	−20.0
			165	−27.0	−21.0	−16.0
H2Xptr	2.88	182	**154**	−14.0	−19.0	−28.0
			137	−14.0	−22.0	−25.0
Bip	3.12	238	**220**	−27.0	−16.0	−23.0
			178	−27.0	−21.0	−19.0
Ptr	3.13	164	**119**	−12.0	−25.0	−22.0
			92	−12.0	−33.0	−16.0

**Table 2 molecules-28-01267-t002:** Precision of the method (*n* = 3) for 7,8-H2Biopterin (H2Bip) and biopterin (Bip) quantification.

Concentration (nM)	RSD (%)	
	H2Bip	Bip
5	3.7	
50	0.6	1.7
100	0.3	0.9
500	0.8	2.4
1000	0.6	0.6
2000	0.9	0.9

**Table 3 molecules-28-01267-t003:** H4B autoxidation kinetics parameters as a function of pH and type of buffer.

		H4Bip				qH2Bip			
pH	Ammonium Buffer	Half-Life (min)	Time Interval (hours)	Reaction Order	Rate Constant (h^−1^)	Half-Life (min)	Time Interval (hours)	Reaction Order	Rate Constant (h^−1^)
2.8	Formate **	87	0–3.4	1	0.5	106	0.8–3.9	1	0.4
		38	3.4–8.7		1.3	38	3.9–9		1.1
3.0	Acetate	25	0–4	1	1.7	34	0–6.6	1	1.2
2.8	Citrate	83	0–3.6	1	0.5	120	0–29.0	1	0.3
5.4	Formate **	66	0–2.3	0	2.5 *	79	0–2.7	0	1.8 *
		15	2.3–3.9	1	3.3	16	2.7–3.9	1	2.7
	Acetate	22	0–3	1	1.9	20	0–3.5	1	2.1
	Citrate	23	0–3	1	1.8	29	0–3	1	1.4
7.4	Formate **	127	0.3–5.7	1	0.3	134	0.3–6.4	1	0.3
		64	5.7–14.3		0.6	74	6.4–14.8		0.6
	Acetate	37	0–6	1	1.1	35	0–6.6	1	1.2
	Citrate	57	0–6	1	0.7	154	0–5	0	0.6 *

* Rate constants for reaction order are expressed in mol^−1^ h^−1^. ** For formate buffer at pH 2.8 and at pH 7.4, where the experiments have been performed in triplicates on three different days, RSD of half-lives of H4Bip and qH2Bip are lower than 14%, in all instances.

**Table 4 molecules-28-01267-t004:** Autoxidation kinetics as a function of H4Bip concentration.

	H4Bip		qH2Bip	
H4Bip Concentration (µM)	Half-Life (min)	Time Interval (hours)	Half-Life (min)	Time Interval (hours)
0.5	85	0.3–3.4	66	0.3–5.8
	53	3.4–5.8		
1.0	193	0.7–5.4	129	0.7–14.3
	107	5.4–14.3		
2.0	299	3.4–12.0	406	3.4–23.3
	245	12.0–23.3		

## Data Availability

Data supporting reported results are available on request.

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
