# Peer review of "Autoxidation Kinetics of Tetrahydrobiopterin—Giving Quinonoid Dihydrobiopterin the Consideration It Deserves"

_molecules, 2023, doi:10.3390/molecules28031267_

Round 1
Reviewer 1 Report
Cofactor metabolites are central to metabolism, however due to the low abundance and low stability, developing analytic chemistry methods to directly assay cofactor intermediates is challenging yet critical. Tetrahydrobiopterin, BH4, is an essential cofactor in aromatic amino acid metabolism, however research on BH4 is lacking partly due to its instability and the difficulty to assay the cofactors directly. Here, this study developed a HPLC -MS/MS method to assay tetrahydrobiopterin cofactors, methodologically optimized different buffer condition to study BH4 oxidative degradation kinetics.
The report might improve a broader impact by presenting as least one application of the developed method to assay BH4 intermediates from any biological samples, cells, tissues or body fluid. For the most optimized buffer condition in vitro, it would be interesting to determine if the degradation kinetics obtained from the biological samples is comparable to the in vitro condition, instead by other enzyme-mediated degradation present in the biological source.
Author Response
Response
First, we would like to express our gratitude to the reviewer for his time in reading this paper and for his valuable comment.
Of course, the reviewer is right about the impact of applying the method to the determination of pterins in biological media. This is why we have indicated in the conclusion section that we are in the process of validating the application of this method to the determination of pterins in biological media.
Indeed, we are in the process of validating the method to assay H4Bip in cerebrospinal fluid (CSF), where the sample preparation procedure consists of a simple filtration to eliminate the pterins followed by a dilution in the mobile phase before injection in the chromatograph (c.f. ref. 30).
The results show that the chromatographic profiles are in all points similar to those observed for pure solutions of H4Bip (Fig. 2 a and b). The validation of the method will of course include the determination of the stability of these pterins in CSF samples, but also the physiological values as a function of age and sex as well as the pathophysiological variations on a sufficient number of CSF samples.
We feel that the reviewer will agree that this is a very substantial work that deserves to be published in a separate paper. Hence, any disclosure of a preliminary result will undermine the novelty of its content. We would therefore be grateful if he could wait until the finalization and publication of this paper, which we will probably submit to Molecules.
Reviewer 2 Report
Reviewer Comments
The authors present a study on “Antioxidation of tetrahydrobiopterin-Giving quinonoid dihydrobiopterin the consideration it deserves.” The study is informative, useful, and relevant to the journal. I have a few revisions and recommendations that need to be addressed by the authors before this manuscript is accepted for publication.
Suggested corrections and revisions
Line 19: This is mainly due to its short half-life of 0.9 to 5 minutes, according to previous studies.
Lines 20 & 21: Revise the sentence…here we developed a HPLC-MS/MS method able to separate qH2Bip from H4Bip and other oxidation products in less than 3.5 minutes.
….here, we developed an efficient HPLC-MS/MS method to achieve the separation of qH2Bip from H4Bip and other oxidation products in less than 3.5 minutes.
Line 46: Delete the word thanks
Lines 77 & 78: revise the sentences….we have demonstrated that H2Bip isomers can be distinguished Molecules 2022, 27, x FOR PEER REVIEW 3 of 14 78 thanks to their MS/MS transitions [37].
we have demonstrated that H2Bip isomers can be identified and distinguished based on their respective MS/MS transition ions [37].
Line 100: …connected to a Shimadzu LCMS-8040 triple quadrupole mass spectrometry.
Line 101, Section 2.11. Liquid Chromatography… The experimental details are not clear to the reader and require additional details on the following…
Did the author use a binary mobile phase system? Gradient or isocratic method? If it is a binary mobile phase system used, what is the mobile phase A? even though the authors have mentioned in the following section that the method is isocratic, the details are required under this section for clarity. Revise this part.
Lines 107 & 108, Section 2.1.2 Revisions are recommended
MS/MS experiments were performed in multiple reaction monitoring (MRM) acquisition modes with positive electrospray ionization (+ESI) source.
Line 139: 3.1.1. Selecting MS/MS conditions.
Replace Selecting with optimization of…
Line 154: 3.1.2. Selecting separation conditions
Replace Selecting with optimization of chromatographic …
Lines 171 & 172: Needs revision
The authors described that the qH2Bip is well separated from H4Bip and H2Bip in this section. The chromatogram clearly shows the separation between qH2Bip and H2Bip but not a significant separation qH2Bip and H4Bip. However, irrespective of their separation, it is possible to accurately determine the inseparable molecules with characteristic fragmental ions using advanced tandem MS.
Line 200: To the best of our knowledge, this is the first time that qH2Bip is separated from? and..
Complete the sentence for better clarity
Lines 209 & 210: It is possible to use an Atlantis T3 column with larger dimensions than that proposed 210 herein in order to increase the efficiency of the separation and thus the resolution.
What do you mean by larger dimensions? If the column length is longer, we can enhance separation capability but requires a longer run time of analysis. Based on the short half-life of the analytes this may not work. If particle sizes are larger, the resolution will be affected. Explain what you want to emphasize under this section in detail.
Lines 215 & 216: Likewise, it is possible to enhance the detection sensitivity by increasing the concentration of methanol at the column outlet in order to improve the yield of ionization in the ESI source.
Do you mean increasing injection volume to enhance ion intensity? Clarify
Lines 259 & 262: For this purpose, we checked the linearity of the proposed method between 5 and 1000 nM for both pterins (y 261 = 0.006x - 0.021, n = 6, and y = 0.005x + 0.057, n = 6, with R2 > 0.99 for H2Bip and Bip, 262 respectively) (Fig. 4). T
Figure 4 displays the concentration range from 5nM to 2000 nM?
Author Response
Suggested corrections and revisions
Response. First, we would like to thank the reviewer very much for his time and especially for his help in improving this paper.
We have made all the corrections suggested by the reviewer. These corrections have been highlighted in yellow in the revised version.
Line 19: This is mainly due to its short half-life of 0.9 to 5 minutes, according to previous studies.
Lines 20 & 21: Revise the sentence…here we developed a HPLC-MS/MS method able to separate qH2Bip from H4Bip and other oxidation products in less than 3.5 minutes.
….here, we developed an efficient HPLC-MS/MS method to achieve the separation of qH2Bip from H4Bip and other oxidation products in less than 3.5 minutes.
Line 46: Delete the word thanks
Lines 77 & 78: revise the sentences….we have demonstrated that H2Bip isomers can be distinguished Molecules 2022, 27, x FOR PEER REVIEW 3 of 14 78 thanks to their MS/MS transitions [37].
we have demonstrated that H2Bip isomers can be identified and distinguished based on their respective MS/MS transition ions [37].
Line 100: …connected to a Shimadzu LCMS-8040 triple quadrupole mass spectrometry.
Response. We changed all the sentences according to the suggestions of the reviewer.
Line 101, Section 2.11. Liquid Chromatography… The experimental details are not clear to the reader and require additional details on the following…
Did the author use a binary mobile phase system? Gradient or isocratic method? If it is a binary mobile phase system used, what is the mobile phase A? even though the authors have mentioned in the following section that the method is isocratic, the details are required under this section for clarity. Revise this part.
Response. L. 102. We added “Isocratic” to Chromatographic separations to clarify this point.
Lines 107 & 108, Section 2.1.2 Revisions are recommended
MS/MS experiments were performed in multiple reaction monitoring (MRM) acquisition modes with positive electrospray ionization (+ESI) source.
Response. We changed the first sentence accordingly.
Line 139: 3.1.1. Selecting MS/MS conditions.
Replace Selecting with optimization of…
Response. It’s done.
Line 154: 3.1.2. Selecting separation conditions
Replace Selecting with optimization of chromatographic …
Response. It’s done
Lines 171 & 172: Needs revision
The authors described that the qH2Bip is well separated from H4Bip and H2Bip in this section. The chromatogram clearly shows the separation between qH2Bip and H2Bip but not a significant separation qH2Bip and H4Bip. However, irrespective of their separation, it is possible to accurately determine the inseparable molecules with characteristic fragmental ions using advanced tandem MS.
Response. This paragraph has been rewritten according to the reviewers' suggestions.
Line 200: To the best of our knowledge, this is the first time that qH2Bip is separated from? and..
Complete the sentence for better clarity
Response. We completed the sentence accordingly (L. 201)
Lines 209 & 210: It is possible to use an Atlantis T3 column with larger dimensions (column length and column diameter) than that proposed 210 herein in order to increase the efficiency of the separation and thus the resolution.
What do you mean by larger dimensions? If the column length is longer, we can enhance separation capability but requires a longer run time of analysis. Based on the short half-life of the analytes this may not work. If particle sizes are larger, the resolution will be affected. Explain what you want to emphasize under this section in detail.
Response. By larger dimensions we mean (length and internal diameter). We added this precision in the text. For the length, the reviewer is right, this will require a longer run-time of analysis, which is not appropriate for short half-lives. For column diameter, in theory this should not affect the efficiency of the separation, but from a practical standpoint, narrower columns usually are less efficient than larger-diameter ones even when using a constant linear velocity (Wall effects and extra-column effects). Of course, peak volumes (peak width in volumetric terms) increase with increasing length and column cross-section, or the square of the change in diameter. This results in proportionally smaller peaks (loss of sensitivity). We modified this section accordingly to detail these points to the reader.
Lines 215 & 216: Likewise, it is possible to enhance the detection sensitivity by increasing the concentration of methanol at the column outlet in order to improve the yield of ionization in the ESI source.
Do you mean increasing injection volume to enhance ion intensity? Clarify
Response. No, it's not the injection volume. It's just to increase the % of methanol in the effluent (Addition of a make-up solvent through a T union). Because the % of methanol in the mobile phase is only 3%, which does not favor the ionization yield in the ESI source.
We have amended the text for clarity
Lines 259 & 262: For this purpose, we checked the linearity of the proposed method between 5 and 1000 nM for both pterins (y 261 = 0.006x - 0.021, n = 6, and y = 0.005x + 0.057, n = 6, with R2 > 0.99 for H2Bip and Bip, 262 respectively) (Fig. 4). T
Figure 4 displays the concentration range from 5nM to 2000 nM?
Response. Yes, it is from 5 to 2000 nM. We have corrected this error. Thank you.
Round 2
Reviewer 1 Report
A validation of the method is needed, in particular the stability of certain pterins from biological samples, because longer qH2Bip half-life appears to be the key finding in the report.
Author Response
Response
We would like to thank the reviewer again for his efforts and insistence on improving the quality of this paper. Yes, he is right, longer qH2Bip half-life is the key finding in the report. We feel that he agrees that the message of this paper is very clear for all those interested in pterins.
Considered together with those we have already published in Anal Chem (Ref. 37), the results of this paper clearly show that whatever the medium and the pH considered, the half-life of qH2Bip is at least equal to that of H4Bip. So, reading this paper, anyone can deduce that this must also apply to biological media, as the reviewer himself has noted. Precisely, the main objective of our work is to pave the way to biological applications. Therefore, we think that adding validation of our results to biological media will not necessarily increase the impact of the paper. This will only undermine the novelty of the paper we are preparing on the LC-MS/MS determination of pterins in biological media.
In our humble opinion, we believe that is rather urgent to publish this paper as is, notably because it contains important information for all specialists in the field, whether chemists, biologists, pharmacists, or physicians. We sincerely hope that the reviewer will accept to share this opinion with us.